# The State of Mapillary: An Exploratory Analysis

**Dawei Ma[1], Hongchao Fan [2,*], Wenwen Li [3] and Xuan Ding[1]**

[1]  School of Remote Sensing and Information Engineering, Wuhan University, Wuhan 430079, China;

[2]  Department of Civil and Environmental Engineering, Norwegian University of Science and Technology, Trondheim 7491, Norway;

[3]  School of Geographical Science and Urban Planning, Arizona State University, Tempe, AZ 875302, USA

**\***  Correspondence:  hongchao.fan@ntnu.no; Tel.: +47-73559665

**Abstract:** As the world's largest crowdsourcing-based street view platform, Mapillary has received considerable attention in both research and practical applications. By February 2019, more than 20,000 users worldwide contributed approximately 6.3 million kilometers of streetscape sequences. In this study, we attempted to get a deep insight into the Mapillary project through an exploratory analysis from the perspective of contributors, including the development of users, the spatiotemporal analysis of active users, the contribution modes (walking, cycling, and driving), and the devices used to contribute. It shows that inequality exists in the distribution of contributed users, similar to that in other volunteered geographic information (VGI) projects. However, the inequality in Mapillary contribution is less than in OpenStreetMap (OSM). Compared to OSM, the other main difference is that the data collection demonstrated obvious seasonal variation because contributions to OSM can be accomplished on a computer, whereas images have to be captured on the streets for Mapillary, and this is considerably affected by seasonal weather.

**Keywords:** Mapillary; volunteered geographic information (VGI); inequality; contribution behaviors

## 1. Introduction

The advanced technology in geographical positioning system (GPS) embedded mobile devices and Web 2.0 allows citizens to participate directly in the construction of geospatial data, promoting the development of volunteered geographic information (VGI) [1]. The VGI data sources include image sharing websites (e.g., Flickr and Panoramio), personal social media platforms (e.g., Weibo and Twitter), and interactive mapping plans (OpenStreetMap and OSM). Mapillary (https://www.mapillary.com), as the first crowdsourcing-based platform to provide detailed street images, allows contributors to capture and upload street-level images through mobile apps, regardless of the travel mode: Walking, cycling, or driving. Besides the supporting app upload provided, web interface, desktop application, and command-line tools enable the flexible upload of images and videos to Mapillary. The convenient contribution methods have prompted volunteers to participate in contribution activities since the beginning of 2014, and now Mapillary has a total of 543.8 million images, according to the official statistics.

Compared to Google Street View (GSV), images on the Mapillary platform follow the Creative Commons Attribution-ShareAlike 4.0 International (CC BY SA 4.0) License, which everyone can use for free. Mapillary images have been widely applied to the construction of street-level datasets for detecting cars, skies, and 64 other object categories to achieve a semantic understanding of street scenes [2]. Since Mapillary images include information about time and geographic coordinates, social and physical behavior of humans can be studied on a worldwide scale through an examination of user contribution patterns [3]. Similar to other VGI data, user contribution patterns and the data

accumulation process are the fundamental differences between Mapillary and traditional street view data platforms [4], and these analyses can provide a basis for answering questions about the quantity, quality, and type of Mapillary data [5, 6]. Walden-Schreiner et al. [7] used Flickr photos to assess seasonal patterns of visitor use in protected mountain areas, providing a basis for the effective management of protected areas. Li et al. [6] conducted an exploratory analysis of the characteristics of the contributors using Twitter and Flickr data. However, related studies on Mapillary are lacking. Juhász and Hochmair [8] analyzed the early stage of Mapillary, considering the overall development and individual contribution behavior of Mapillary in the first year, such as days of active contributions. As of now, Mapillary has been under development for more than five years, but no recent research has examined its development status and user contribution patterns.

Analysis of user contribution behaviors can help us better know the data sources, and the data sources are closely related to the Mapillary data quality [8, 9]. In this study, we did an exploratory analysis of Mapillary data in order to have a deep insight into the Mapillary community. The analysis of changes in the number and contribution of users in different regions can help us assess user loyalty in data collection. In general, a user-friendly platform is attractive to users; Mapillary in particular provides a multi-mechanism data sharing opportunity. We assume that users continue to contribute images to Mapillary, which is significantly responsive to the timeliness requirements of street view data, while many high-end street view platforms (like GSV) data updates take a long time and a huge amount of resources. In addition, knowing about the user's contribution patterns (walking, cycling, and driving) and the equipment used for contribution can give us a more targeted understanding of the data composition of Mapillary. Contribution inequality has substantial and complicated impacts on data quality and on the developments of the project [4]. We expect the emergence of a small set of users who do most of the work in the Mapillary community. That means that most data come from active contributors with expertise in the tools used and rich experiences, which may lead to higher data quality, but also means that a small number of contributors could have a huge impact on the project. Therefore, the analysis of these active users is more representative for us to explore the contribution behaviors in Mapillary.

The rest of the paper is structured as follows: Section 2 reviews the related work. Section 3 introduces the data sources and the current status of Mapillary. Section 4 explores the imbalances in Mapillary. The contribution behaviors of major users are analyzed in Section 5. Finally, the results of this paper are summarized in Section 6.

## 2. Related Work

The contribution inequality problem is common in VGI data contributions, with research reported in Wikipedia [10], Flickr [3], Panoramio [3], and OSM [5]. Arazy and Nov [11] explored both the direct and indirect effects of contribution inequalities in Wikipedia and found that global inequality significantly positively impacts article quality. Yang et al. [4] conducted a temporal analysis of the inequality of contribution in OSM in different countries, indicating that contribution inequality is related to the import of data. After exploring the emergence of participation inequality on both temporal and spatial scales and evaluating the implications for the use of VGI, Haklay [12] proposed that the contribution inequality in the analysis of a VGI project must be considered. In the initial Mapillary research, contribution inequality was also found [8], showing that a small number of people have a large average contribution per week, accompanied by a longer contribution activity. User contribution to Mapillary has been ongoing. How will the inequality of contributions in Mapillary change? Analyzing the changes in contribution inequality helps in understanding the development of Mapillary. This was explored in this study.

The quality of VGI data has always been the focus of attention. Many studies have evaluated the data quality of OSM from different object levels such as road network [13], point [14], line [15], and polygonal [16]. Hagenauer and Helbich [17] mentioned, in their research, that nearly all 'empirical studies have shown that urban areas are better mapped' in OSM . Data completeness is one of the major data quality elements, Juhász and Hochmair [8] analyzed the completeness of Mapillary and believed that its data have better coverage on pedestrian and cycle paths in some cities than GSV.

Vladimir et al. [18] used Mapillary and GSV images to determine the geographic location of the detected object, and experiments showed that, as the amount of Mapillary images increased, the detection accuracy was closer to that of high-end street data. In many scenarios, Mapillary can already be used as a substitute for GSV images to some extent. The development of VGI has produced large amounts of related research, not only related to the quality of VGI data, but also regarding contributors who have participated in VGI activities. Neis and Zipf [5] divided OSM contributors into four categories according to the number of nodes contributed by members and analyzed the attributes of members, such as location and activity area. Alivand and Hochmair [3] analyzed the photo contribution patterns of Panoramio and Flickr in California, USA, including the type of data growth and the environment associated with data contribution counts. Yang et al. [19] relied on a behavior-based approach to assess the expertise of major contributors and judged whether they were professionals or amateurs. Juhász and Hochmair [8] analyzed the Mapillary user contribution patterns on both the country and continental level, as well as the individual level, revealing the early development of Mapillary. The analysis of the contribution patterns in VGI projects helps in further exploring people's behavior mechanisms and in evaluating the quality of VGI data. Antoniou and Schlieder [20] studied the spatial behavior of contributors to OpenStreetMap in the Greater London Area, linked it to gamification mechanisms, and found that it can help in addressing participation issues with spatial allocation games. Zielstra et al. [21] proposed a method to determine the home regions of the OSM contributors, based solely on the spatial clustering of the created nodes by analyzing the OSM data types contributed and the editing behaviors of users. Bégin et al. [22] suggested studying contributors' mapping behavior, such as preferences for objects to map, to understand the characteristics and quality of the data produced.

## 3. Overview of Mapillary Today

In this section, Mapillary data are analyzed statistically to provide an overview of the development in Mapillary to date.

### 3.1. Data Preparation

Mapillary data contain images captured by GPS-embedded devices when contributors are driving cars, riding bicycles, or walking. When the Mapillary app is used, the device automatically captures street view images at regular intervals. Thus, each image can be represented by a GPS node with a timestamp. These consecutive nodes form a trajectory, which is called a sequence in Mapillary. If the app is stopped or the GPS signal is lost, then the subsequently acquired images are considered as another sequence.

Mapillary data can be accessed through its official Application Programming Interface (API), including images and sequences as the two main types. In this article, we use the sequence data for analysis. We did not consider the image content, and only accessed and downloaded the attribute information of the sequence. The data cover the period from January 2014 to February 2019. Mapillary allows developers to search for data using different Uniform Resource Locator (URL) parameters (Table 1), and we use the "bbox" and "end_time" as URL parameters to access data. "Bbox" means filtering by the bounding box, defined by four parameters, including the minimum/maximum longitude and latitude of the bounding box. "end_time" represents filtering sequences that are captured before the end time. Since Mapillary only has 1000 data per page for a particularly dense area, it was necessary to hierarchically partition the bounding box for iterative queries in order to obtain complete sequence data. We divided the bounding box containing more than 1000 sequence records into four equal-size blocks, and then queried and downloaded the data of each small block. A block was further divided until no sub-block contained more than 1000 sequences.

**Table 1.** Uniform Resource Locator (URL) Parameters used to search for data in Mapillary.

| URL Parameter | Type | Description |
|---|---|---|
| bbox | Number[] | Filter by the bounding box, given as minx, miny, maxx, maxy. |
| closeto | Number[] | Filter by a location that images are close to, given as longitude, latitude. |
| end_time | Date | Filter images that are captured before end_time. |
| image_keys | Key[] | Filter images by a list of image keys. |
| lookat | Number[] | Filter images that images are taken in the direction of the specified location (and therefore that location is likely to be visible in the images), given as longitude,latitude. Note that If lookat is provided without geospatial filters like closeto or bbox, then it will search global images that look at the point. |
| organization_keys | Key[] | Filter images by organizations. |
| pano | Boolean | Filer panoramic images (true) or not (false). |
| per_page | Number | The number of images per page (default 200, and maximum 1000). |
| private | Boolean | Filter images by private (true) or public (false). |
| radius | Number | Filter images within the radius around the closeto location (default 100 meters). |
| sequence_keys | Key[] | Filter images by sequences. |
| start_time | Date | Filter images that are captured since start_time. |
| userkeys | Key[] | Filter images captured by users, given as user keys. |
| usernames | String[] | Filter images captured by users, given as usernames. |

The downloaded sequence data were in Geojson format. In total, we acquired about 37 GB of data. For subsequent spatial operations and analysis, we stored the sequence data in a PostgreSQL database with PostGIS extensions. Each sequence had various properties, as shown in Table 2. Each user corresponds to a separate "user_key", and each sequence also has its own separate "key". In case of image blur and/or GPS signal loss, Mapillary prevents the device from taking images and closes the current sequence. However, this mechanism was not implemented in the early days of Mapillary. To remove these data errors in the earlier version, Juhász and Hochmair [8] removed straight line segments that were 1 km long or longer. Under normal circumstances, if the Mapillary app has a signal and works normally, the distance between the two data points uploaded by the user should be relatively close. If the two points are far apart, this indicates that the app is likely experiencing an error. In this study, we also used 1 km as the threshold to remove erroneous data to ensure that the sequences represented the true coverage of the captured images.

**Table 2.** Sequence properties in Mapillary.

| Property | Type | Description |
|---|---|---|
| camera_make | String | Camera model name |
| captured_at | Date | When sequence was captured |
| created_at | Date | When sequence was uploaded |
| coordinateProperties | Object | Properties for coordinates |
| key | Key | Sequence key |
| pano | Boolean | Whether the sequence is panorama (true), or not (false) |
| user_key | Key | User who captured the sequence |
| username | String | Username of the corresponding user. |

On Mapillary, no information is provided about where a user is located, because users do not have to provide geographical profiles (i.e., country of residence) during registration or data upload. Therefore, identifying the country that a Mapillary user is from is difficult. In this study, we identified users' countries of residence based on the common knowledge that users normally contribute in the areas where they live. The rules used for the identification were as follows:

(1) Calculate the number of days the user contributes in different places, and the place with the largest number of contributions determines the user's country;

(2) When the contribution days are the same, the user's country is determined according to the sequence length of the cumulative contribution;

(3) When neither of the above two was able to produce an identification, the user's country was determined according to the location of the user's first contributed sequence.

It is difficult to exactly identify the user's residence area. In general, the longer a user contributes in a country, the more likely he or she is living here. Therefore, we first determined the country of residence based on the number of days that the user contributed. The length of the sequence indicates the time and extent of the user's contribution in the area. Usually, local users are more likely to make long-distance streetscape contributions. When neither of the current rules are available, we assume that the first sequence created by the user is located in close proximity to his or her residence. In fact, new users generally create their first new objects in areas that they are familiar with [5]. Notably, the country identification of users in this work does not mean nationality or the country to which they belong in any political sense. In this work, country identification is where users are living when they contribute to Mapillary, if they are identified with a country.

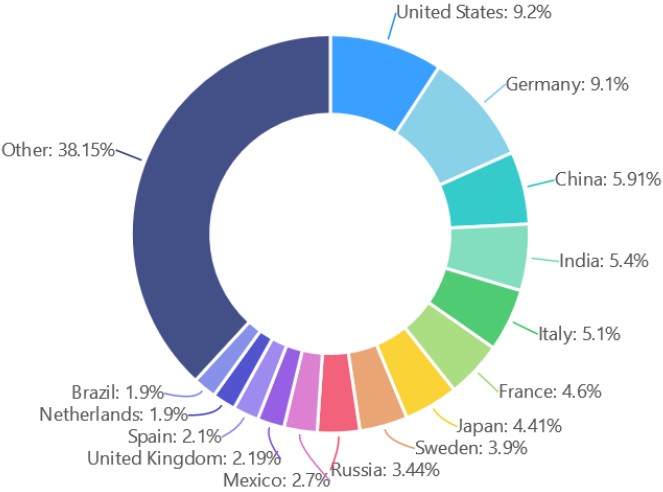

**Figure 1.** The distribution of users in various countries.

Figure 1 depicts the distribution of Mapillary users by country based on the above rules. Only the top 14 countries are shown here. The statistical results indicated that all identified users live in 190 countries. We were unable to determine country information for 2.63% of users, maybe due to factors such as GPS positioning offset. By the end of February 2019, about 9.2% of the total users were identified with the United States, having the largest number of users in all countries. A significant number of users were concentrated in European countries. Among the top countries were Germany, France, Sweden, and the United Kingdom. Similar results were also reported in the OSM statistics, where the number of German members was also at the forefront [5]. This reflects the VGI project having a good development environment in Germany, and many people recognize the VGI development pattern and are willing to participate in the activities. Sweden, as the origin of the Mapillary project, also has a certain user base. Apart from the United States, the countries with a large number of users in the Americas are Mexico and Brazil. We found many users from China and India in Asia.

### 3.2. Overall Mapillary Situation

In total, by February 2019, there were 2,609,215 sequences for the whole world, with a total length of 6,256,943.445 km. On some roads, the sequences contributed by users were overlapping. These data were contributed by 21,948 users and cover 211 countries and regions. Figure 2 depicts the monthly contribution in terms of the length of sequences around the world. A certain number of street view image sequences were uploaded to Mapillary each month since 2014, and the total length increases each year. In Figure 2, the user's highest monthly sequence length contributions are concentrated in July or August of each year (the most contributions in 2018 were provided in May, and the contributions in July and August were slightly lower). This is the time when users contribute the most each year. The trough corresponding to the annual contribution in Figure 2 occurs in January or February of each year. However, the overall observations, the corresponding peaks, troughs, and mean values continue to grow each year, following a linear tendency. Simultaneously, the distribution of peaks and troughs is similar every year, showing a certain regularity, which indicates that the contribution of users is related to time. We discuss this in further detail in Section 5.1.

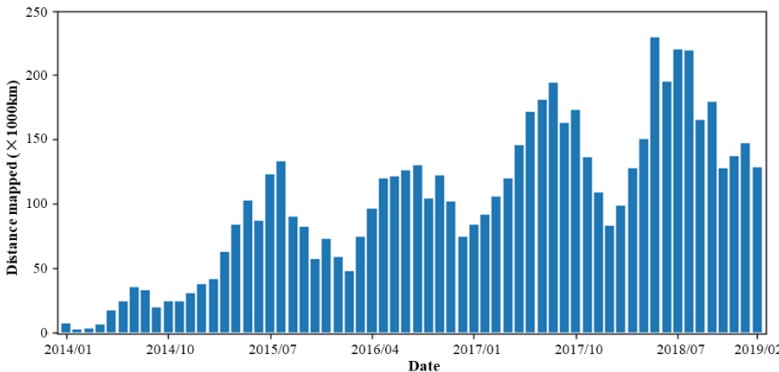

**Figure 2.** The length of sequences contributed by users each month.

To explore the differences in the development of Mapillary in different regions, we analyzed the changes in both the global sequence data and the number of users at the continent level. Almost no activity was recorded in Antarctica and thus it was not considered. The lines in Figure 3 show how the total length of the Mapillary sequences collected in each continent changed monthly from January 2014 to February 2019. These histograms show the cumulative number of users participating in the Mapillary contribution activities for different years on each continent from 2014 to 2018. While the growth rates in the sequence length of various continents are different, they all show a gradual growth trend. As of February 2019, the total length of sequences in Europe, North America, and Asia ranked in the top three. Among them, Europe and North America far exceeded the length of the sequence in Asia, indicating that the users in these two regions are active, and the contribution to Mapillary is positive. The growth of Mapillary data is inseparable from the constant participation

and contribution of new and old users. By the end of 2018, the number of users in Europe was far ahead, followed by Asia and North America. The number of users on the other three continents was relatively small. The growth rate of European users far exceeded that of other continents. Considering the sequence length of each continent, in general, the development of Mapillary has been faster in Europe. A large number of users participate in Europe, and the overall amount of data is large. In comparison, both the total length and the growing rate of the sequences in North America are significantly higher than those in Asia, whereas the number of active users in Asia is higher than in North American every year. This means that active users in North America are more productive than those in Asia.

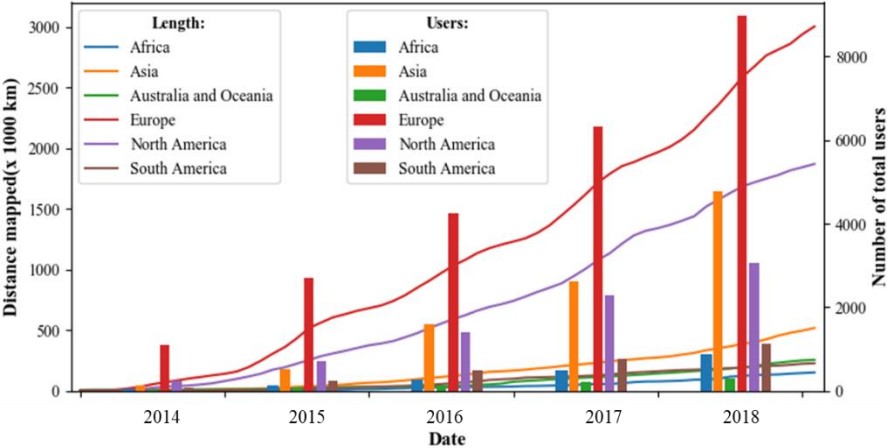

**Figure 3.** Variations in the sequence length and number of users in each continent.

We divided contributors into six parts, based on the year of contribution to Mapillary since 2014. Figure 4 depicts the monthly composition changes in the type of contributors. A significant number of new users join the Mapillary data collection activity every year, and some old users still contributed. Some users who have contributed since the beginning of the project in 2014 are still active in 2019.

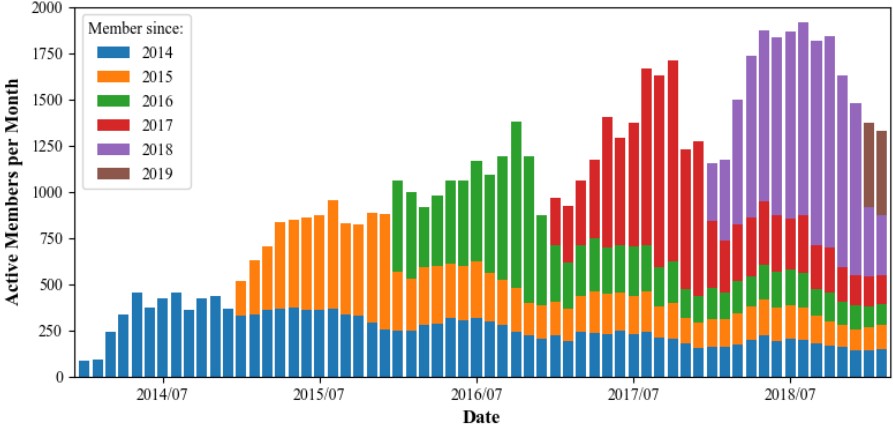

**Figure 4.** Changes in the composition of users.

We counted the changes in the number of users who continued to contribute uninterruptedly each year. In Table 3, the first number in each row indicates the number of new users who started to contribute in year A (the listed year for that row or column). The following cell provides the number of users who still contributed in year B (the year listed in the corresponding column). For example, 1740 users started to contribute in 2014, 779 of which still contributed in 2015. By February 2019, 140 of these old users participated in the contribution for more than five years. These users are mainly from 40 countries around the world. The specific distribution is displayed in Figure 5. The largest number of users was in Germany, where 20 users contributed Mapillary data every year for more

than five years, followed by the United States and Japan, with 14 and 11 users, respectively. According to the statistics, the top three users of the total length of the contribution are among the 140 old users. The user with the longest contributed sequences is from the United States, with a cumulative contribution of more than 278,000 km. The user contributing the second-most is also from the U.S. and the third is from Sweden.

**Table 3.** Number of users who continue to contribute each year.

| Year | 2014 | 2015 | 2016 | 2017 | 2018 | February 2019 |
|------|------|------|------|------|------|---------------|
| 2014 | 1740 | 779 | 539 | 376 | 273 | 140 |
| 2015 | - | 2924 | 876 | 455 | 299 | 115 |
| 2016 | - | - | 4068 | 1299 | 391 | 118 |
| 2017 | - | - | - | 5190 | 1084 | 205 |
| 2018 | - | - | - | - | 7205 | 513 |

**Figure 5.** The distribution of users who have contributed every year from 2014 to February 2019.

While the older users continue contributing for many years, the number of new users is increasing every year. The rapid growth of the community indicates that increasingly more people are supporting Mapillary, guaranteeing the follow-up development and data enrichment of Mapillary. The Mapillary users who have provided long-term contributions often have more experience and collection techniques, which, to some extent, ensure that Mapillary receives better quality data. The continuous contribution activities show that Mapillary provides a good user experience that is attractive to users. Overall, the Mapillary community is in a good state.

We statistically analyzed the length of the Mapillary sequence in countries. The results showed that the development is unbalanced between countries; the value in a small proportion of a region is very high, whereas the sequence length in many regions that occupy a large area of the world map is short. To more clearly and intuitively describe the length of the unit sequence of Mapillary in each region, we used an algorithm [23] to display the data (Figure 6). Figure 6 shows that regions with a higher sequence length per unit area are mainly concentrated in Europe. The Netherlands ranks as number one in the world, followed by Germany, Belgium, Denmark, and other European countries. In North America, the length of the sequence per unit area is higher in the United States. In Asia, many countries have relatively low values, so countries such as Russia and China, which occupy a large geographical area, are strongly distorted. Only Japan and Thailand have relatively high values. Many users in China and India are involved in Mapillary, but, as seen in Figure 6, the length of the sequence within the unit area of these two countries is small. Many users may simply attempt a single Mapillary data collection or contribute on a small scale. For China, Mapillary data are mainly

concentrated in Taiwan and Hong Kong. Overall, Mapillary's global coverage is uneven. Most European countries and the United States have more abundant data.

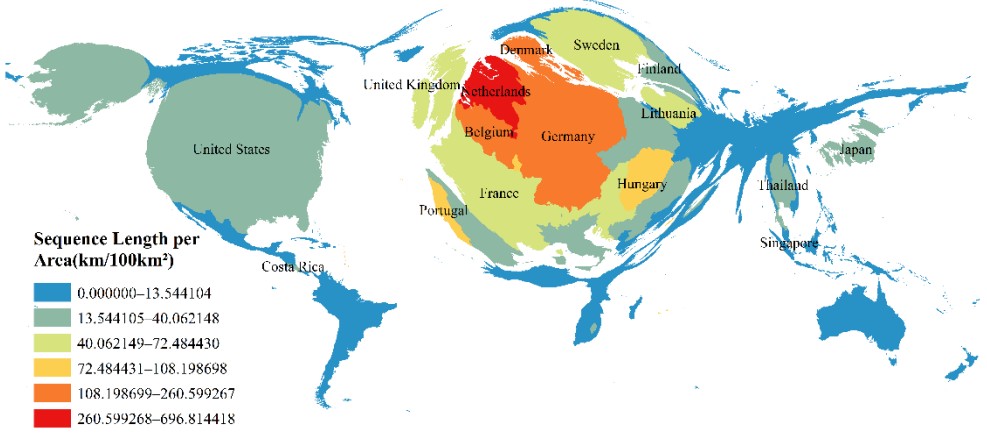

**Figure 6.** Country/region unit area (100 km²) sequence length.

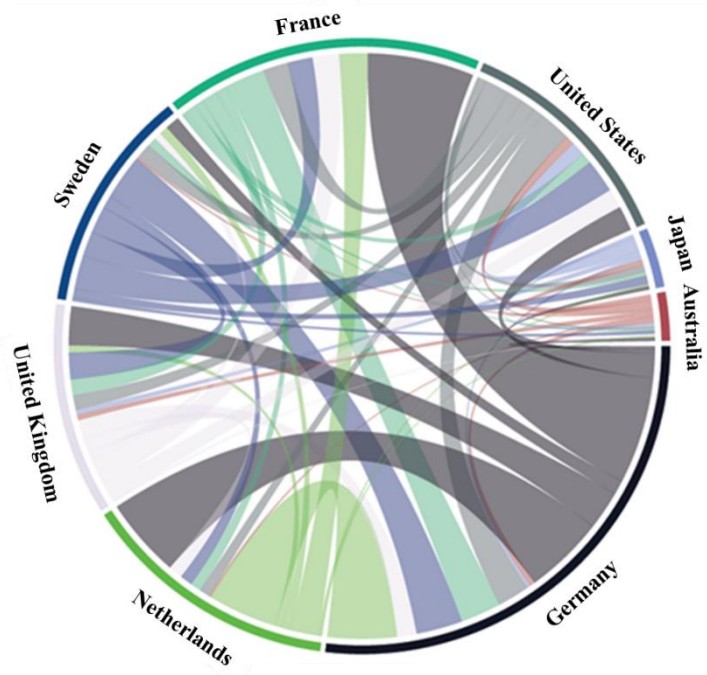

**Figure 7.** User composition in selected countries.

We selected countries with a large number of users and rich sequence data to explore the country composition of users. Figure 7 shows the mutual contribution of users from eight countries, including Germany, the United States, and France. The different countries in the figure are represented by arcs of different colors, and the bands of the same color as the arcs indicate that the users of the corresponding countries have contributed to other countries. The wider the ribbon, the more users are represented. Users contributing in foreign countries is common in Mapillary. Users in countries with close geographical proximity contribute more to each other. Many German users have contributed to Mapillary in other countries. Many sequences collected by these users are found in many European countries such as the Netherlands and the United Kingdom. Most German users contributed to France. Many users in other European countries have also mapped Mapillary sequences in Germany. Users in the United States also have close links to the European countries and they contributed to each other. The user composition in Japan and Australia is diverse, but the number of users connected to the other six countries is relatively low.

From mobile phones to high-end cameras, the Mapillary platform can process images from a variety of devices. Therefore, the tools used by users for data collection are diverse. Figure 8 shows

the distribution of devices used by contributors when collecting data. Here, we only display devices with more than 1000 sequences. Among them, Samsung, Apple, Microsoft, Garmin, and Sony are the leaders. Capture settings include smartphones (Samsung, Huawei, Apple, etc.) as well as professional capture devices (Trimble, Garmin, etc.). Smartphones of various brands are the most widely used by users. The devices available are versatile and the acquisition environment is not limited, which is extremely user-friendly. People can use simple tools to collect street view images and contribute to Mapillary, which facilitates data collection for users and enhances the contribution experience of new users.

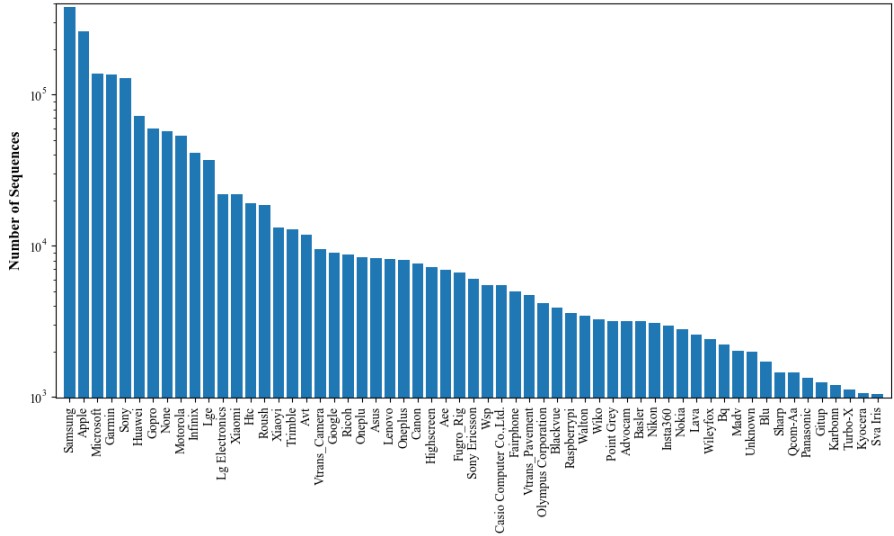

**Figure 8.** Distribution of camera sensors/devices used for sequence capturing.

The camera sensors and devices used in each country to collect Mapillary sequences are diverse. We counted the top five equipment most used to capture Mapillary sequences in some countries, and found that 16 main types were used in the nine countries (Figure 9). The base graph in Figure 9 shows the number of street view images per unit area (1 km²) using the same algorithm as Figure 6. The distribution of the number of images per unit area is similar to the sequence length within the unit area (Figure 6), and Europe has a higher image density. We selected some countries with high density to count the more frequently used (top five) device types and present the results in donut charts. Apple was the most commonly used brand, and ranked among the top five in all the selected nine countries. Samsung and Sony were also popular.

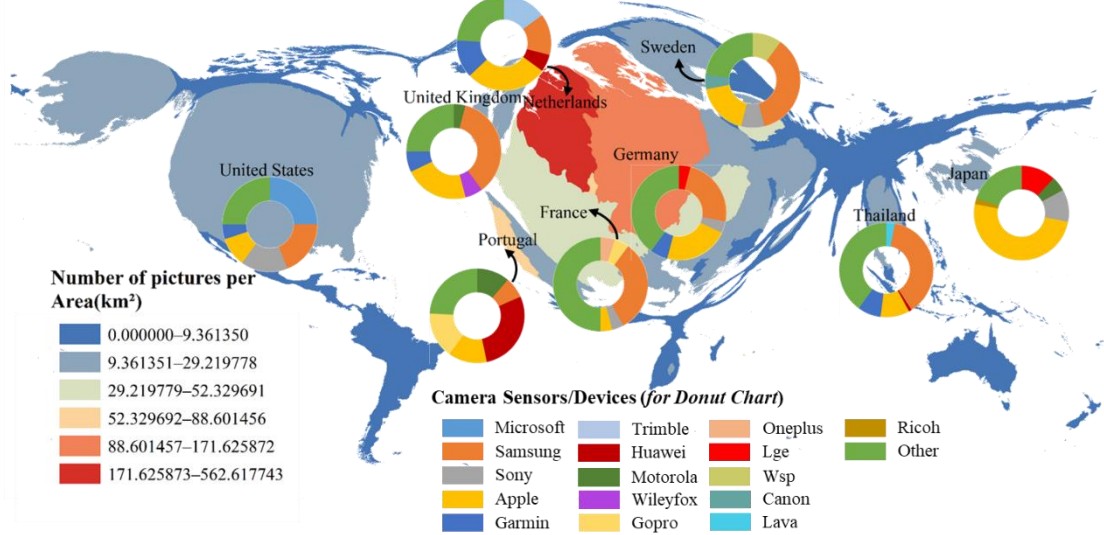

**Figure 9.** Top five camera sensors/devices most used in some countries. The base map is the number of images per unit area (km²) for each country.

Mapillary allows people to collect street view data while walking, cycling, or driving. Various contribution models help people to better participate in Mapillary. We calculated the average speed of each sequence to determine the contribution method when the user was collecting the data. To reduce the impact of equipment and positioning errors, we removed the sequences with fewer than 20 images. For the remaining sequences, we obtained the time when the first and last images in the sequence were captured, and used the interval between them as the total collection time. The ratio between the geographical distance corresponding to the sequence and the total time was taken as the average speed. According to the information obtained, the walking speed of most people was 5 km/h, whereas cycling speed was about 20 km/h. In this study, sequences with a speed below 8 km//h were assumed to be collected while walking. Sequences with speeds greater than 25 km/h were considered to be collected by driving. The other sequences were captured by the users while cycling. Figure 10 shows the distribution of the number of sequences and images captured in different ways. The sequences with less than 20 images accounted for 40.9%, but the corresponding total number of images was very small, accounting for only 0.5% of all images. This is consistent with the relationship between sequence length and the number of images. That is, the shorter the sequence, the fewer the images. A large number of short sequences indicate that most contributors have only made simple attempts. In Figure 10, 72.8% of the images were collected by driving. This means that most users prefer to use Mapillary when driving. Generally, users travel further and more easily by driving, so that the number of images in distant trajectories was also large. This further explains why the images included in the sequences contributed by driving accounted for 72.8% of all images.

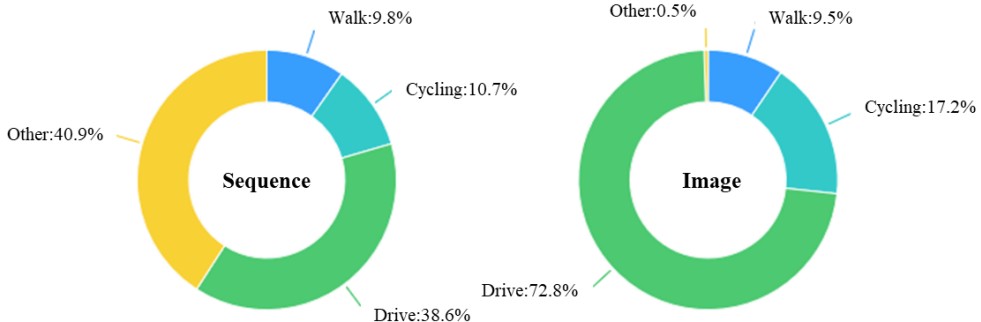

**Figure 10.** Distribution of contribution modes used for capturing data.

## 4. Contribution Inequality

Figure 11 depicts the distribution of the number of sequences and images contributed by users from high to low. The histograms in Figures 11a and 11c are highly right-skewed, and this distribution state is consistent with the form of the power law. The power law is often called a heavy tail or long tail distribution, indicating an imbalance in the quantity distribution [24]. In this paper, it indicates that many Mapillary users contributed a small number of images and sequences, whereas a small number of users contributed a large amount of data. Figures 11b and 11d are the corresponding log-log plots. As the number of sequences and images increases exponentially, the number of users who provide these contributions significantly decreases. Juhász and Hochmair [8] also found that Mapillary users have similar distribution patterns in terms of average weekly contribution distance, active days, and radius of gyration.

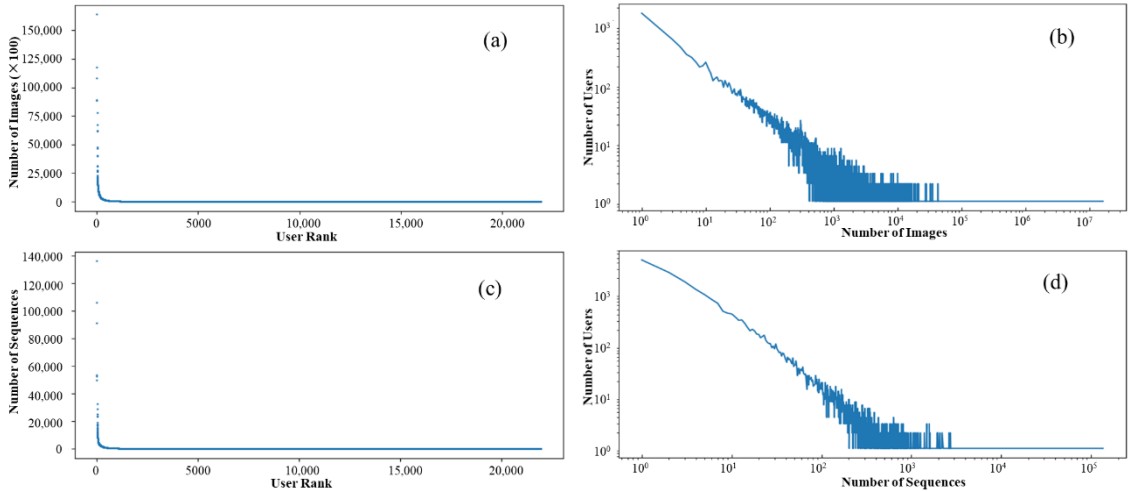

**Figure 11.** Distribution of the **(a, b)** number of images and **(c, d)** sequences; **(b, d)** log–log plots.

The user contributions in Mapillary are unbalanced, so we wanted to determine how the inequality in the Mapillary community changed over time and if any differences existed between different regions. Yang et al. [4] proposed a method of quantifying inequality to explore temporal changes in participation inequality in OSM contribution activities. In this study, we also used the Gini coefficient (*G*) to depict a quantitative measure of contribution inequality. *G* is a single number that measures the degree of inequality in a distribution. Its value ranges from zero to one, where a larger value indicates a higher level of inequality. *G* is usually defined mathematically, based on the Lorenz curve. As shown in Figure 12, we sorted the contribution of the users in ascending order, and set the cumulative share of contributors as X and the cumulative share of contributions as Y in order to obtain the Lorenz curve of the Mapillary contributions. The line at 45 degrees, which is called the line of equality, represents perfect equality of contributions (the value of *G* is zero). *G* can then be thought of as the ratio of the area that lies between the line of equality and the Lorenz curve (*A*) over the total area under the line of equality (*A* and *B*). That is:

$$G = \frac{A_{Area}}{A_{Area} + B_{Area}}.$$

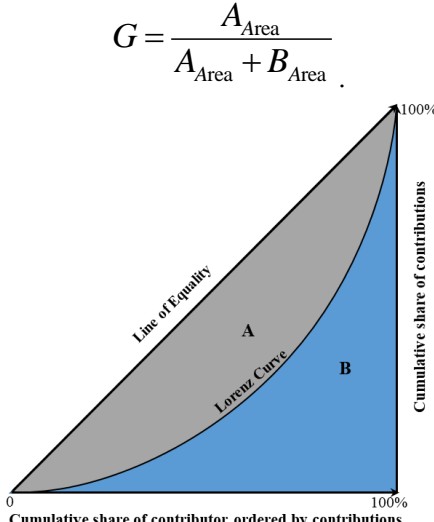

**Figure 12.** The concept of the Gini coefficient (A represents the gray part of the graph, and B represents the blue part.).

To further explore the time and space differences in user contribution inequalities, we selected some countries with abundant data and extensive distribution in order to analyze the variation in *G*. Figure 13 shows that *G* values of the seven countries are relatively high, whereas the contribution

inequality from 2014 to 2018 fluctuated, but the value increased. This shows that extreme contribution inequality exists in all countries and the situation is still growing. In 2018, *G* of all countries exceeded 0.9, of which Japan was far higher than 0.95. These values are significantly higher than Wikipedia's contribution inequality (0.84) [25], but most are less than OSM (more than 0.95) [4].

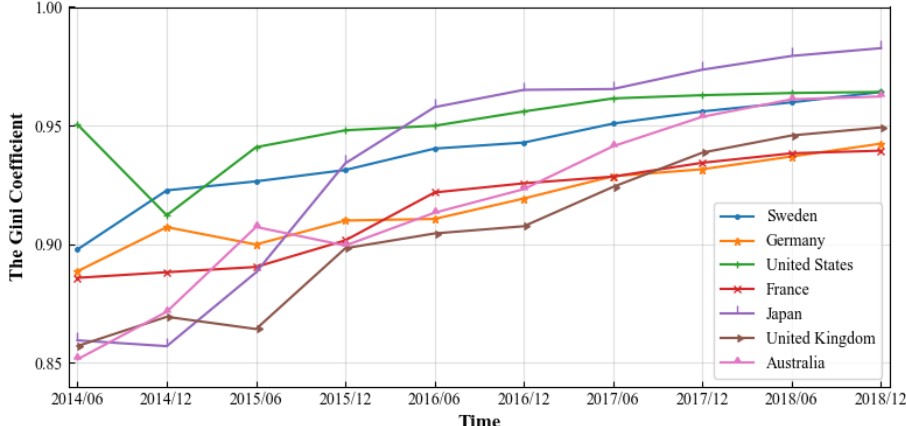

**Figure 13.** The Gini coefficient of selected countries.

We also calculated the indicator of the top X% of all contributors, accounting for Y% of all contributions to obtain more information in order to further illustrate the changes in contribution inequality by country. Figure 14 shows the changes in the selected countries every three months from 2014 to 2019, including the percentage of top users who completed 95%, 90%, 80%, and 50% of all contributions. The proportion of top users required to complete the same contribution has declined significantly over time because most of the contributions of new volunteers are relatively few, and a small number of old users have contributed a large amount of data in the past and continuously collect images, so the cumulative contribution ratio increases. In Germany, for example, providing 90% of contributions in March 2014 required about 40% of users, whereas by 2018, only 13% were required. Notably, users in Australia, the United Kingdom, and other countries in early 2014, due to the relatively small number of users, 90% or more contributions required all the users. However, over time, the difference in contributions between users increases and the number of users required to complete these contributions drops dramatically.

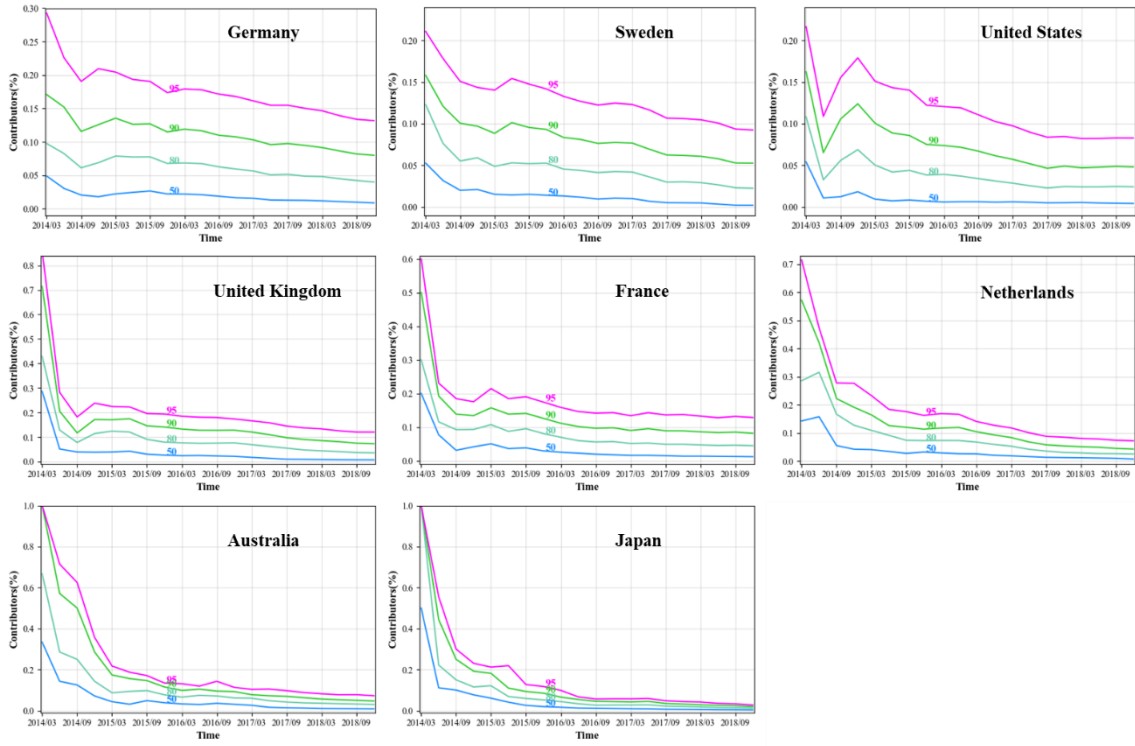

**Figure 14.** The percentage of contributors to reach a certain percentage of contributions.

## 5. Behaviors of Major Contributors

A large number of users contribute only a relatively small amount of data, and the majority of contributed data is the result of the top users' work, so the behavior of a small number of users can reflect the Mapillary community more objectively. In this study, we chose users with relatively large contributions as the main contributors to the analysis, primarily including top users who have provided more than 90% of the contributions (the length of all sequences) from 2014 to 2019. We called them "major contributors". According to statistics, a total of 1294 users were selected. These users come from 111 countries. Table 4 shows the top 15 countries with the total number of major contributors. Among them, the United States and Germany have the largest number, with 222 and 143, respectively.

**Table 4.** Top 15 countries with the total number of major contributors.

| Country/Region | Number of Users |
|---|---|
| United States | 222 |
| Germany | 143 |
| France | 83 |
| Sweden | 56 |
| Russia | 43 |
| Brazil | 37 |
| Poland | 36 |
| United Kingdom | 35 |

| Country/Region | Number of Users |
|---|---|
| Netherlands | 34 |
| Italy | 30 |
| Spain | 30 |
| Thailand | 26 |
| Denmark | 25 |
| Japan | 25 |
| Ukraine | 25 |

The data contributed by these major contributors have a wide geographical distribution. We calculated the proportion of each country in terms of the number of sequences and the length of the sequence in the total contributions as of 2019. Table 5 shows the results of user contributions in the top 15 countries and is ordered by the number of sequences. The major contributors in the US contributed nearly 30% of the total sequences length and the total number of sequences exceeded 26%, both of which far exceeded those of other countries. Similar to Table 4, the number of sequences contributed by these users in Germany ranks second. We further analyzed the contribution patterns of these major contributors.

**Table 5.** Top 15 countries with the total contributions (ordered by number of sequences).

| Country/Region | Number of Sequences (%) | Length of Sequences (%) |
|---|---|---|
| United States | 26.16% | 29.85% |
| Germany | 8.71% | 10.56% |
| Australia | 7.78% | 4.60% |
| Sweden | 6.93% | 5.13% |
| France | 6.87% | 5.70% |
| Japan | 4.73% | 2.24% |
| Netherlands | 3.65% | 5.09% |
| Russia | 3.30% | 2.10% |
| Romania | 2.73% | 2.77% |
| Mexico | 2.02% | 0.87% |
| Italy | 1.94% | 1.35% |
| Belgium | 1.80% | 1.42% |
| Brazil | 1.79% | 1.73% |
| Denmark | 1.61% | 1.96% |
| Thailand | 1.54% | 1.64% |

*5.1. Temporal Pattern of Contributions*

Each sequence has a "captured_at" attribute (Table 1) that represents the time at which the

corresponding street view sequence was captured. This attribute helps us explore the time distribution of the user's contribution activities. Due to the large time zone differences and the size of the global region, we selected some countries from Table 5 with different time zones and geographical locations for comparative analysis. Figure 15 shows the contributions of each country in different months. Sweden, the Netherlands, and Germany are located in the higher latitudes of the Northern hemisphere. The contribution trends of Germany and the Netherlands are similar for each month: Gradually rising from January to May and declining from August to the end of the year. Contributions in June and July were significantly lower compared to May and August. Sweden, which lies at a higher latitude than the other two countries, had a higher contribution in June, July, and August. This is likely mostly related to temperature and the length of daytime in different seasons. In hot weather in Germany in June and July, people reduce their outdoor activities and their contribution to Mapillary decreases accordingly. At other times, as the monthly temperature changes, people's contributions continue to change accordingly. Due to the high latitudes in Sweden, the temperature is not too high, but it is relatively warm during June, July, and August, and the main contributors are more active during those months. United States and Japan, which have lower latitudes, have no such obvious trends. Corresponding to Australia in the Southern hemisphere, the main contribution months were concentrated in November, December, and January, as the temperature is higher throughout the year.

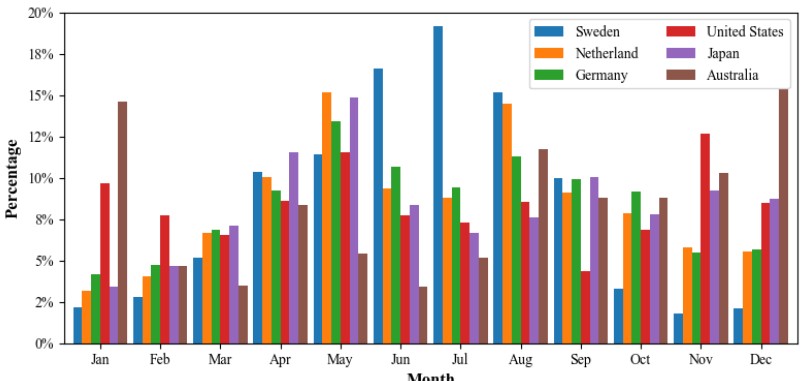

**Figure 15.** Monthly contribution of users.

Mapillary uses UTC format to describe the time attribute of the sequence. We converted UTC to the corresponding local time to explore the user's contribution at different periods during the day. Here, we used the above six countries as examples for analysis. For countries with a wide range of longitudes, such as the United States and Australia, we used the country's central time zone as the basis. Figure 16 depicts the time distribution of the major contributor acquisition sequences. User contributions are obviously tied to the time of day. Many contributions are provided during the noon period, and few contributions were made in the morning and evening. While the timing of the peak level of contributions each day varies from country to country, the overall trends are similar: They all rise gradually before the highest contribution and then fall. Both the rise and fall are accompanied by fluctuations, which is in line with our general understanding. The shooting of street view data requires certain lighting conditions, and better data can be obtained during the day. Japan and Australia still provided relatively high contributions in the middle of the night. We observed some corresponding images content and found that the property time of many images were wrong. We noticed that the image time of the Mapillary platform states that it is late at night, but the image content is of daytime scenes. This shows that a problem occurred with the time attribute of a batch of data in the Mapillary data, which may be caused by user upload, by an error when performing the UTC conversion.

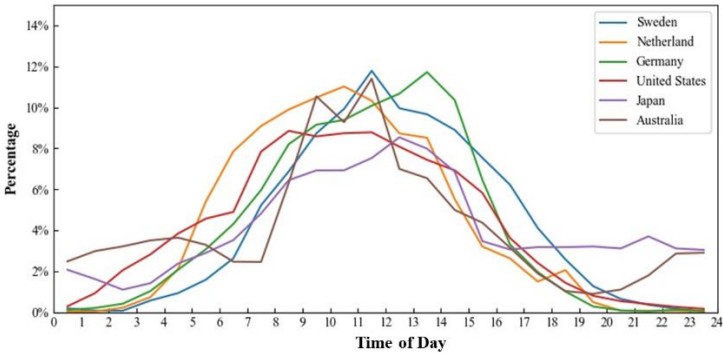

**Figure 16.** Users' contribution time distribution.

*5.2. Spatial Contribution Density of Major Users*

The Mapillary contributors show a phenomenon of repeated contributions on one road. To analyze the repeated contribution of major users, we defined spatial density as an indicator of quantitative analysis. Figure 17 shows the processing steps for calculating spatial density. First, the minimum bounding rectangle was obtained, based on all sequence positions contributed by the user. Then, we divided the minimum boundary rectangle into grids of 500 × 500 m (grids with edges less than 500 m were extended). Subsequently, the grids that did not intersect the sequences were removed, and the remaining grids represented the region where the user provided contributions to Mapillary. These remaining grids are called "contributions grids" in this paper. Finally, we calculated the number of sequences that intersected with each on the contribution grids. This quantity is the spatial density of the corresponding grid. The higher the density, the more frequently the user is active in the area, and the street view data in the area are relatively rich. Figure 18 visualizes the partial spatial density of one of the major users. The user comes from Bangladesh and the area indicated in the figure is located in Dhaka. The darker the color, the more the sequences in the grids.

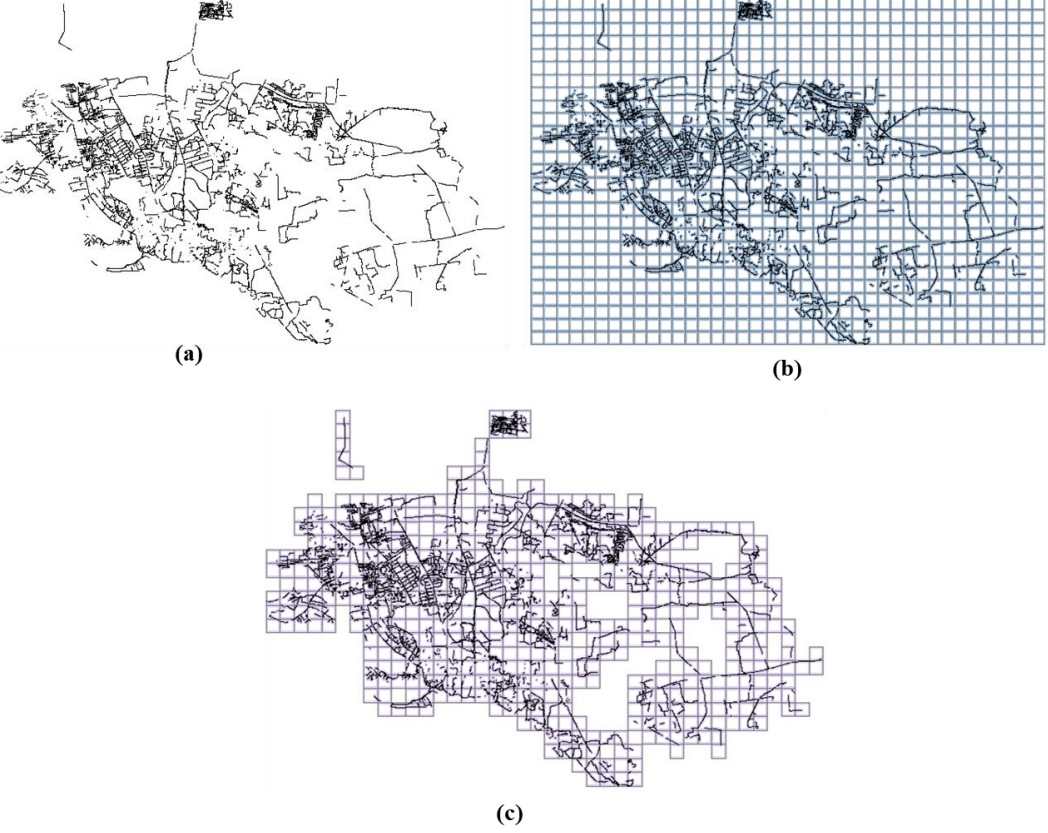

**Figure 17.** Spatial density calculation process: (**a**) Sequences, (**b**) grids generated based on the minimum bounding rectangle, and (**c**) actual contribution area.

**Figure 18.** An example of user's spatial contribution density. The base map is the OSM map somewhere in Dhaka. The darker the color, the more the sequences in the grids of 500 × 500 m.

When different users contribute to Mapillary, the areas of activity vary considerably. Therefore, the number of contribution grids corresponding to different users is also different. In the analysis, users were divided into three categories according to the number of grids. The proportion of users with a contribution area covering less than 1000 grids was 30.3%, which we called small-scale contributors; the medium-scale contributors, whose cumulative grid numbers were between 1000 and 2000, occupied about 44.7% of the total users; and the proportion of large-scale contributors with cumulative grid numbers exceeding 2000 was 25.0%.

Patterns with a measure value above given threshold are called interesting patterns with respect to that measure [26]. Similarly, for all contribution grids we defined a grid with a density greater than three as the active grid. An active grid means that users contribute at least four sequences in the grid, indicating that users have passed through multiple times in the corresponding area. These areas are usually relatively familiar or important to the user. In general, small-scale contributors are more likely to repeatedly contribute data in a certain area. Statistics also showed that the proportion of active grids of the three groups of users accounted for 24.4%, 12.3%, and 8.9% of the total contributions of each group, respectively. This indicates that, as the scope of user contributions expands, the proportion of areas where users repeatedly contribute data decreases. The area in which a person is constantly active is limited. When users contribute data in a wider range, many places are new to these users, which are passed by only occasionally, so the spatial density of many grids will be less than three.

Figure 19 displays the distribution of the number of active grids for each group of users. As the scope of contributions expands, the proportion of users with more than 200 active grids increases significantly. The proportions are 29.3%, 37.7%, and 53.7%, for small, medium, and large-scale contributors, respectively. When expanding the scope of contribution, users inevitably provide repeated contributions in the previously contributed regions. This may occur because the user's living environment and the surrounding road system are relatively fixed. Therefore, the user must repeatedly pass through some traffic and living areas when contributing street view data to new areas. However, the new contribution area is still much larger than the area of these repeated contribution areas. Thus, as the area of total contribution increases, the proportion of the area corresponding to the active grids decreases from 24.4% to 8.9%.

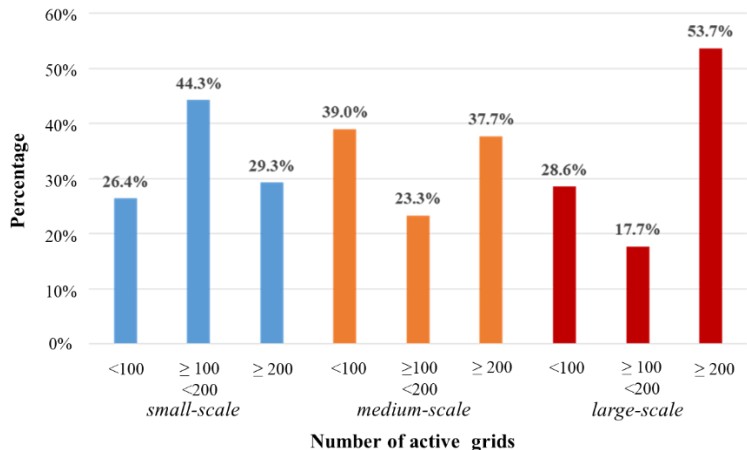

**Figure 19.** Number of active grids per contributor group.

## 6. Conclusions and Future Work

In this study, we analyzed the overall situation of Mapillary from many aspects. According to the data collected, by February 2019, a total of 21,948 users worldwide had contributed to Mapillary. Among them, 140 users continuously contributed data from the beginning of 2014 to February 2019. Mapillary data vary widely by continent. Street view images are abundant in Europe and North America with many users around the world. We determined the user's country according to set rules. About 62% of users were from Europe, Asia, and North America. These users were mainly identified with the United States, Germany, China, India, and Italy. We identified 2019 users in the United States, the largest number of users in all countries. For all users, approximately 10% contributed to at least two countries. Users in countries with geographical proximity provide more mutual contributions. We found large differences in sequence length and the number of images in different regions. Many European countries and the United States had long Mapillary sequences and a large number of street view images. While many users were located in Asia, the overall sequence length was short. This was shown in the sequence length distribution within the unit area of each country. Only Japan and Thailand had a certain length in Asia, indicating that users in Europe and North America were more productive than those in Asia. The devices that users use when collecting data were diverse. Frequently used capture settings included smartphones (Samsung, Huawei, Apple, etc.) and professional capture devices (Tianbao, Garmin, etc.). Users can collect data using hand-held devices while walking, and can also contribute to Mapillary when cycling and driving. More than 70% of street view images were captured by users while driving. These analyses give us a further insight into the composition of Mapillary data from different perspectives. Meanwhile, our research reveals the true source of crowdsourced data and the quantitative distribution of Mapillary, which can help us understand the development of Mapillary at different stages.

Similar to other VGI data contribution platforms, we found inequalities in the contributions in Mapillary. A small percentage of users contributed most of the street view data, which indicated that Mapillary's data quality was closely related to a small number of users. We selected 1294 major users for analysis who were top users that contributed more than 90% of the cumulative sequence length from 2014 to 2019. We mainly analyzed the main users from two aspects: Time and space. The statistical results showed that the user contributions were closely related to temperature and daytime length in different seasons. Users contributed more data during the warmer months. The contributions at noon were the highest. During the hot months, people reduced their outings and their contributions decreased. We explored the repeated contributions of major users in certain spatial areas. The concept of spatial density was proposed for quantitative analysis. According to the scope of the users' activities, we divided users into small, medium, and large-scale contributors. We used areas with a spatial density greater than three to represent where the user is most often active and explored the distribution of active grids for different groups of users. The results showed that, as the

scope of contributions expands, the scale of repetition increases. However, the proportion of the repeated area was much smaller than the area where users newly captured the street view images. This reflected that the contributions of most users were always around a specific area, which can be a place of residence or office space. These few contributors have a huge impact on the project and can most intuitively reflect the contribution behaviors of the Mapillary community.

The exploratory analysis in this paper provides a more intuitive understanding of the development process of the Mapillary community and they provide a valuable basis for answering questions about Mapillary's data quality [5]. The Mapillary data also include some other properties, such as the "edits" attribute enabling the user's further data processing. The user can be subsequently analyzed by combining more attributes with the quality of the user's images. In addition, we can aggregate contributors from different locations with similar contribution behaviors to analyze the group behavior characteristics of contributors [27].

To date, passionate users (users who keep contributing since registration) of Mapillary increases steadily by more than 100 annually. This guarantees the healthy development of the project. In the next few years, images on Mapillary will continue to increase rapidly, and the problem of inequality in contributions will remain. However, the inequality in image coverage will significantly decrease, because people will contribute to less covered regions. At the same time, we believe that Mapillary will draw more attention from the research domain, especially with urban geomatics and autonomous driving.

**Author Contributions:** Hongchao Fan contributed toward creating the original ideas of the paper and designed the experiments. Dawei Ma and Xuan Ding prepared the original data. Dawei Ma performed the experiments and analyzed the experimental data under the supervision of Hongchao Fan. Dawei Ma wrote the first draft of the manuscript, while Hongchao Fan and Wenwen Li revised and edited it. All authors have read and agreed to the published version of the manuscript.

**Funding:** This work is supported by the NSFC (National Natural Science Foundation of China) project No. 41771484.

**Acknowledgments:**

**Conflicts of interest:** The authors declare no conflict of interest.

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
