# Peer review of "The State of Mapillary: An Exploratory Analysis"

_ijgi, doi:10.3390/ijgi9010010_

Round 1

Reviewer 1 Report

I assume this is a review paper, but it is more like a blog article with poor English here and there. I didn't see any novel methods or conclusion from the manuscript. Most analysis are not in-depth for a journal article. The pattern of contribution behavior in Mapillary is kind of similar to other vgi or citizen science projects. I don't really understand why the authors submitted such manuscript to a journal, probably a blog or website fits better to your content.

Author Response

Thanks a lot for reviewing the paper. This is not a review paper. We may argue that we did valuable work for the research community. And this is a serious scientific contribution rather than a blog article. BTW, we asked colleagues of native English speaker to polish the language and do not think that the language is that bad as you mentioned.

This paper did an exploratory analysis to help readers having a deep insight of Mapillary project. Specifically, we did the spatiotemporal analysis of active users, found out the contribution modes (walking, cycling, and driving), and the devices used to contribute. We also figured out the inequality of the contribution in Mapillary. Base on the analysis, we make a prediction for the development and applications of the project. We do think that this means a lot for the research community.

Reviewer 2 Report

This paper gives an overview of the global trends of Mapillary, which has been attracting attention as an example of VGI in recent years.

Based on such reviews, reviewer hope that case studies and various analyzes will be deepened in the future works.

In adopting this paper, it is desirable to consider the following contents as minor revisions.

1. As for the reviews of previous studies, many studies in the field of Computer Vision have analyzed the data of Mapillary as well as VGI. I would like to quote and explain some of these.

2. l137: It is difficult to specify the user's residence area. Focusing on data uploads, there are many users who use only at travel destinations, but it is better to explain how you thought about the algorithm.

3. l269: Regarding the analysis of Wordcloud, since quantitative evaluation is performed by region in Figure 10, it may be necessary to show only the total count in the text.

4. l365: Because the range of the vertical axis (Contributors) is different for Figure 15, there is a possibility of giving the reader a different interpretation when placed side by side. Is there a room for rethinking how to represent figures?

5. l386: Regarding the analysis of Figures 16 and 17, the trend is considered to be different within the same country depending on the main contributor's contribution region. Can you provide a breakdown of the 1294 users' countries and the main posting areas targeted this time? Also, because Figure 16 is fine, the meaning cannot be read, so there is room for reconsideration.

Author Response

This paper gives an overview of the global trends of Mapillary, which has been attracting attention as an example of VGI in recent years. Based on such reviews, reviewer hope that case studies and various analyzes will be deepened in the future works.

In adopting this paper, it is desirable to consider the following contents as minor revisions.

As for the reviews of previous studies, many studies in the field of Computer Vision have analyzed the data of Mapillary as well as VGI. I would like to quote and explain some of these.

Reply: Thank you for your advice. The quality of VGI data has always been the focus of attention, and we cited some related researchers from line 94 to 104.

l137: It is difficult to specify the user's residence area. Focusing on data uploads, there are many users who use only at travel destinations, but it is better to explain how you thought about the algorithm.

Reply: Thank you for your advice. Due to the lack of detailed user information, it is true that accurate determination of  user's residence is difficult. In this paper, Country identification is where a user stays when she or he contributes to Mapillary. We set some rules to identify users’ countries of residence mainly based on some common knowledge.

In order to make our rules easier to understand, we added some brief explanations at line 176-182.

l269: Regarding the analysis of Wordcloud, since quantitative evaluation is performed by region in Figure 10, it may be necessary to show only the total count in the text.

Reply: Thank you for your advice. The use of Wordcloud is intended to show the use of the camera device more intuitively, but it does overlap with Figure 8 in the content. We have removed this figure.

l365: Because the range of the vertical axis (Contributors) is different for Figure 15, there is a possibility of giving the reader a different interpretation when placed side by side. Is there a room for rethinking how to represent figures?

Reply: Thank you for your suggestion. We adjusted the Figure from two rows to three rows, and placed the figures with similar range of the vertical axis in the same line. It could be easier for reader to compare.

l386: Regarding the analysis of Figures 16 and 17, the trend is considered to be different within the same country depending on the main contributor's contribution region. Can you provide a breakdown of the 1294 users' countries and the main posting areas targeted this time? Also, because Figure 16 is fine, the meaning cannot be read, so there is room for reconsideration.

Reply: These 1249 users come from 111 countries. The United States and Germany have the most users, with 222 and 143 respectively. The data contributed by these major contributors also have a wide geographical distribution, and the amount of data varies greatly from country to country.  We added some brief explanations and two tables (Table 4 and Table 5) to explain the breakdown of the 1294 users.

Reviewer 3 Report

It is an interesting approach. It is a well written paper. I am a little bit confused why do you use the words images, pictures and photos? Is there a difference? Row 255 it is not correct you should write Many German users or many users in Germany. In general you must leave some space after the figures. For example row 366 the next section no. 5 begins immediately. Some figures must be improved. Figure 7 seems to be a sophisticated figure but it is not clear. Figure 9 it is a nice design but not well understood. Figure 10 too small fonts they are not readable. Figure 19 it is very small.

Author Response

It is an interesting approach. It is a well written paper. I am a little bit confused why do you use the words images, pictures and photos? Is there a difference?

Reply: Thank you for pointing this out. There is no difference between them and they are all used to represent Mapillary images (pictures, photos) uploaded by contributors. We modified it and uniformly used "image".

Row 255 it is not correct you should write Many German users or many users in Germany.

Reply: Thank you for reviewing our paper carefully. We are very sorry for our incorrect writing, and we have modified it to “Many German users”.

In general you must leave some space after the figures. For example row 366 the next section no. 5 begins immediately.

Reply: Thank you for your advice. We have added some space after our figures.

Some figures must be improved. Figure 7 seems to be a sophisticated figure but it is not clear. Figure 9 it is a nice design but not well understood. Figure 10 too small fonts they are not readable. Figure 19 it is very small.

Reply: Thank you for your suggestion. (1) We regenerated Figure 7 and modified the font of the image content to make it clearer. (2) Figure 9 (Wordcloud) is used to indicate how many times various devices are being used by contributors. The larger the font, the more frequent the device used. But the figure is somewhat repetitive in the content with Figure 8, so we removed it as suggested by other reviewers. (3) Since there are legends for the donut charts in Figure10, we decided to remove the small words you mentioned (if the font is larger, it will affect the display of the donut charts). The adjusted figure does not affect the reader's access to information. (4) We adjusted Figure 19 into two lines and enlarged the figures (now it is named Figure 17).

Reviewer 4 Report

Overall, I enjoyed reading and reviewing this paper. This type of study for Mapilliary is timely. The paper is well written, the structure and style are good. The English is of a high standard. The paper is potentially publishable.

The introduction of the paper very briefly outlines that the authors will analyze the contribution behaviours of the contributors to Mapillary. The contribution of the major users are analysed. This is a weakness of the paper. There is no real set of objectives outlined here rather than the paper will perform a very wide and deep exploratory analysis of the contribution patterns. This type of exploratory analysis is fine but it must be guided in some way. Perhaps a set of hypotheses could be used? For example - "We expect the emergence of a small set of users who do most of the work" (better phrased of course).

The paper is very figure heavy. I don't think all of these diagrams or figures are useful or enhance the paper.

Figure 6 and Figure 10 (cartograms) could easily be removed and replaced with tables.

Figure 7 - if a reader prints this in black-and-white (as I did), the colors make no sense. I actually find these charts confusing.

Figure 9 - word clouds are not my favourite and I think they are find when considering spoken word or generated text. I would remove this.

Figure 16 - I would remove this. These are very difficult to read and particularly poor when printed on black-and-white.

Figure 20  - the caption should have more detail without needing to read into the text.

Section 4 is an important section in the paper. This probably stands out as the section with the most scientific merit in the paper.

In the conclusions section I can see a number of specific conclusions: Mapillary data varies widely by continent, the devices that users use when collecting data were diverse, there were inequalities found in the contribution patterns, and the main users were considered from two ways (time and space).

As I have said above (and mention again further down) is the absence of key messages and results from this work. This exploratory study should deliver a number of key findings which essentially provide a "State of the Mapillary 2019" - these can then be referred back to by other researches in coming years. 

The major negative of the paper for me is the positive work of the deep exploratory analysis is negated by the lack of specific outcomes (or the evaluation of specific objectives of the study).

I checked two options above which require some further details

Does the introduction provide sufficient background and include all relevant references?

>> The references are fine. My concern is that there could be a clearer formulation of the actual objectives of the paper.

Are the conclusions supported by the results?

>> There are many results in the paper. However, without a specific set of objectives outlined there is a lack of structure in the results and the discussion. There is a lack of specific "take away messages" from this paper.

Author Response

Overall, I enjoyed reading and reviewing this paper. This type of study for Mapilliary is timely. The paper is well written, the structure and style are good. The English is of a high standard. The paper is potentially publishable.

The introduction of the paper very briefly outlines that the authors will analyze the contribution behaviours of the contributors to Mapillary. The contribution of the major users are analysed. This is a weakness of the paper. There is no real set of objectives outlined here rather than the paper will perform a very wide and deep exploratory analysis of the contribution patterns. This type of exploratory analysis is fine but it must be guided in some way. Perhaps a set of hypotheses could be used? For example - "We expect the emergence of a small set of users who do most of the work" (better phrased of course).

Reply: Thank you for your advice. Analysis of user contribution behaviors can help us better know the data sources, and the data sources are closely related to the Mapillary data quality. The analysis of changes in the number and contribution of users in different regions can help us assess user loyalty in data collection. In addition, know about the user’s contribution patterns (walking, cycling, and driving) and equipment used for contribution can give us a more targeted understanding of the data composition of Mapillary. While contribution inequality means that a small number of contributors could have a huge impact on the project, so the analysis of these active users is more representative for us to explore the contribution behaviors in Mapillary.

We add some explanations from line 57 to 74 to illustrate the real set of objectives of the paper.

The paper is very figure heavy. I don't think all of these diagrams or figures are useful or enhance the paper.

Figure 6 and Figure 10 (cartograms) could easily be removed and replaced with tables.

Reply: Figure 6 and Figure 10(cartograms) have been constructed by algorithm, they are to show the data of some countries in Europe more clearly, but they are still geographical maps in a certain sense and still maintain a certain spatial relationship. Therefore, we believe that more information can be conveyed to the reader than the table graph. So we hope to keep these two figures and remove the other ones to solve the problem of figure heavy.

Figure 7 - if a reader prints this in black-and-white (as I did), the colors make no sense. I actually find these charts confusing.

Reply: Since the flow of people in multiple countries involves the interrelationship between multiple target objects, it is relatively complicated to visualize them. Figure 7 (chord diagram) represents different countries in different colors, and the width of the strip represents the number of people. Color is therefore essential for the acquisition of information. So we hope to keep this figure and remove the other ones to solve the problem of figure heavy.

Figure 9 - word clouds are not my favourite and I think they are find when considering spoken word or generated text. I would remove this.

Reply: Thank you for your advice. We removed this Figure of wordcloud.

Figure 16 - I would remove this. These are very difficult to read and particularly poor when printed on black-and-white.

Reply: Thank you for your advice. We removed this Figure of wordcloud.

Figure 20 - the caption should have more detail without needing to read into the text.

Reply: Thank you for your advice. We add some explanations in the caption.

Section 4 is an important section in the paper. This probably stands out as the section with the most scientific merit in the paper.

In the conclusions section I can see a number of specific conclusions: Mapillary data varies widely by continent, the devices that users use when collecting data were diverse, there were inequalities found in the contribution patterns, and the main users were considered from two ways (time and space).

As I have said above (and mention again further down) is the absence of key messages and results from this work. This exploratory study should deliver a number of key findings which essentially provide a "State of the Mapillary 2019" - these can then be referred back to by other researches in coming years.

Reply: thank you very much for this comment. Indeed, we have done an exploratory study to give the readers an overview and deep insight of what happened and is happening in Mapillary. Therefore, we changed the title and the objective of this paper based on your comment.

The major negative of the paper for me is the positive work of the deep exploratory analysis is negated by the lack of specific outcomes (or the evaluation of specific objectives of the study).

I checked two options above which require some further details

Does the introduction provide sufficient background and include all relevant references?

>> The references are fine. My concern is that there could be a clearer formulation of the actual objectives of the paper.

Reply: Thank you for your advice. As replied in the first question, we add some explanations from line 57 to 74 to illustrate the real set of objectives of the paper.

Are the conclusions supported by the results?

>> There are many results in the paper. However, without a specific set of objectives outlined there is a lack of structure in the results and the discussion. There is a lack of specific "take away messages" from this paper.

Reply: we revised the conclusion and add a short paragraph as take-away message for the readers.